# Functional Prediction of *trans*-Prenyltransferases Reveals the Distribution of GFPPSs in Species beyond the Brassicaceae Clade

**DOI:** 10.3390/ijms23169471

**Published:** 2022-08-22

**Authors:** Jing Zhang, Yihua Ma, Qingwen Chen, Mingxia Yang, Deyu Feng, Fei Zhou, Guodong Wang, Chengyuan Wang

**Affiliations:** 1The Center for Microbes, Development and Health, CAS Key Laboratory of Molecular Virology and Immunology, Institute Pasteur of Shanghai, Chinese Academy of Sciences, Shanghai 200031, China; 2State Key Laboratory of Plant Genomics, Institute of Genetics and Developmental Biology, The Innovative Academy of Seed Design, Chinese Academy of Sciences, Beijing 100101, China; 3University of Chinese Academy of Sciences, Beijing 100039, China; 4State Key Laboratory of Pharmaceutical Biotechnology, School of Life Sciences, Nanjing University, Nanjing 210023, China

**Keywords:** terpenoid, prenyltransferase, product length determination, geranylfarnesyl pyrophosphate, sesterterpenes

## Abstract

Terpenoids are the most diverse class of plant primary and specialized metabolites, and *trans*-prenyltransferases (*trans*-PTs) are the first branch point to synthesize precursors of various chain lengths for further metabolism. Whereas the catalytic mechanism of the enzyme is known, there is no reliable method for precisely predicting the functions of *trans*-PTs. With the exponentially increasing number of available *trans*-PTs genes in public databases, an in silico functional prediction method for this gene family is urgently needed. Here, we present PTS-Pre, a web tool developed on the basis of the “three floors” model, which shows an overall 86% prediction accuracy for 141 experimentally determined *trans*-PTs. The method was further validated by in vitro enzyme assays for randomly selected *trans*-PTs. In addition, using this method, we identified nine new GFPPSs from different plants which are beyond the previously reported Brassicaceae clade, suggesting these genes may have occurred via convergent evolution and are more likely lineage-specific. The high accuracy of our blind prediction validated by enzymatic assays suggests that PTS-Pre provides a convenient and reliable method for genome-wide functional prediction of *trans*-PTs enzymes and will surely benefit the elucidation and metabolic engineering of terpenoid biosynthetic pathways.

## 1. Introduction

With more than 80,000 molecules that have been discovered to date, terpenoids constitute one of the largest and most diverse groups in naturally occurring organic compounds [1]. These various chemicals are of significant interest to humans because of their extensive applications in the food, pharmaceutical, cosmetic, and agriculture industries. Despite their vast structural and functional diversity, terpenoids are de novo originated from two fundamental five-carbon (C5) precursors, namely, isopentenyl pyrophosphate (IPP) and its isomer dimethylallyl pyrophosphate (DMAPP) [2]. These C5 compounds are generated from one of two separate pathways: the mevalonate (MVA) pathway and the 2-C-methyl-d-erythritol 4-phosphate (MEP) pathway [3,4]. Prenyltransferases (PTs, also called isoprenyl diphosphate synthases, IDS) are the critical enzymes that subsequently condense DMAPP and IPP to polyprenyl pyrophosphates with various chain lengths. These series of polyprenyl pyrophosphates, including geranyl pyrophosphate (GPP, C10), farnesyl pyrophosphate (FPP, C15), geranylgeranyl pyrophosphate (GGPP, C20), geranylfarnesyl pyrophosphate (GFPP, C25), and polyprenyl pyrophosphate with higher molecular weight, can be further catalyzed by a group of enzymes called terpene synthases (TPSs) to generate final products of monoterpenes (C10), sesquiterpenes (C15), diterpenes (C20), sesterterpenes (C25), and longer-chain-length terpenoids, respectively (Figure 1) [5,6,7]. Thus, prenyltransferases represent the first branch point and largely determine the terpenoid diversity.

The PTs can be divided into two structurally and evolutionarily distinct classes as either *cis*- or *trans*-PTs according to the carbon–carbon double-bond configuration of the products, and the vast majority of terpenoids are derived from the products of *trans*-PTs [8]. According to the chain length of their predominant products, *trans*-PTs can be further classified as short-chain (*trans*-SC-PTs, C10–C25), medium-chain (*trans*-MC-PTs, C30–C35), and long-chain (*trans*-LC-PTs, C40–C50) PTs [9]. The catalytic mechanisms and protein structures of *trans*-PTs have been extensively explored in the previous studies: *trans*-PTs share a similar overall structure that 10 α helices (A to J) form a large central cavity pocket with two conserved aspartate-rich motifs, DDx2–4D (the first aspartate-rich motif, FARM) and DDxxD (the second aspartate-rich motif, SARM) sited opposite in it. The condensation reactions occur in the cavity pocket, and the elongated polyprenyl groups scrunch into a product chain elongation tunnel constituted by α helices D, E, and F (Figure 1 and Appendix A) [9,10].

The functional detection methods for *trans*-PTs are well established. The products can be either radioactively labeled and analyzed via chromatography or directly detected via HPLC–MS/MS [11,12]. However, these methods are usually time-consuming and expensive, and they may need special permits. On the other hand, it is difficult to perform the functional annotations of uncharacterized *trans*-PTs sequences by simply using the multiple sequence alignment with functionally characterized *trans*-PTs because of the fact that the predominant product chain length may change dramatically upon replacement of a single amino acid [10,13]. Moreover, with the fast development of genome sequencing, the number of uncharacterized *trans*-PT sequences in the National Center for Biotechnology Information (NCBI) protein database has increased exponentially. Accordingly, in silico function prediction methods for *trans*-PTs are urgently needed, and several efforts have been made through different approaches. One method in a previous study tried to make predictions of chain length specificity using a ligand-docking method that evaluates the steric complementarity between different polyprenyl pyrophosphate products and the elongation tunnel [14]. The prediction accuracy of this method is around 60% which is higher than TrEMBL (an automated annotation method from UniProt Knowledgebase/Translated EMBL Nucleotide Sequence Data Library, with 45% prediction accuracy) [15]. However, the method using the structural modeling and docking algorithm requires huge computational resources and may hardly be applied in the annotation for such a large number of sequences from the *trans*-PTs family.

In our previous work, by solving the crystal structures of three different *trans*-PTs from *Arabidopsis thaliana*, we revealed the product chain length determination mechanism of *trans*-SC-PTs and interpreted it as a “three floors” model [10]. Using this model, we successfully identified a novel GFPPS clade distributed in Brassicaceae plants which formed GFPPS–TPS gene clusters in their genomes [10]. These findings finally led to the discovery of a series of sesterterpenoids in plants [16,17,18,19]. However, it is unclear whether the “three floors” model could also be applied to other species, and whether there are more GFPPSs beyond the Brassicaceae clade in plants is unknown. Thus, in this study, we designed an automatic algorithm named PTS-Pre (Prenyltransferases Prediction tool) for predicting the function of *trans*-PTs using the updated “three floors” model. This program gave 86% prediction accuracy for an experimental dataset including 141 *trans*-PTs from Archaea, Bacteria, and Eukarya and achieved the best prediction performance in Eukarya (95% prediction accuracy). The program was also tested by blind prediction using randomly selected *trans*-PTs sequences which were verified in subsequent in vitro enzymatic assays. All these results indicate that PTS-Pre could be extended to other species and could achieve not only high accuracy but also fast (several seconds per sequence) predictions. We further carried out a systematic search of 7870 *trans*-PTs sequences, spanning all domains of life, to mine the potential GFPPS genes. Finally, nine uncharacterized *trans*-PTs from Nostocaceae, Malvaceae, Musaceae, and Ranunculaceae families were predicted and confirmed with GFPPS activities suggesting the evolutionary lineage specific for GFPPS. In summary, our work sheds light on the genome-wide functional prediction of *trans*-PTs and would be of further benefit to terpenoid metabolism pathway elucidation and production engineering.

## 2. Results

### 2.1. Data Collection and Establishment of PTS-Pre for the Prediction of trans-Prenyltransferases

In our previous work, we used the “three floors” model to interpret the product chain length determination mechanism of *trans*-SC-PTs and revealed that the condensation reaction pockets in these enzymes only have enough space for GPP (C10) or FPP (C15), and the longer linear polyprenyl pyrophosphates must scrunch into an additional product chain elongation tunnel formed by three α helices (helix D, E, and F) [10]. The residues from each of the α helices, whose side-chains face toward the center of the tunnel, form the “floors”, and the distance between each “floor” is 5.4 Å (a pitch of α helix), which is similar to the length of a single prenyl group (C5, 5.2 Å). The sizes of the “floor” residues determine whether the prenyl group could pass through the “floor” or would be terminated. Thus, each “floor” from the top to bottom eventually determines the product specificity as FPP, GGPP, GFPP, and PPP (≥C30), respectively (Figure 1). The “three floors” model was further verified by site-directed mutagenesis and applied in the identification of new GFPPSs in the Brassicaceae clade [10].

With the above achievements, however, in the previous study, all the “floor” residue positions from uncharacterized enzymes were located by manually generating sequence alignment through searching template enzymes from *Arabidopsis*. Moreover, the “blocking efficiency” of the “floor” residues was simply classified into three classes as large/medium/small side-chains, while the elongation impedimental influence between the floors was not taken into consideration. Thus, the prediction accuracy may decrease when the target protein has low sequence similarity with the searched templates from *Arabidopsis* and the method can hardly be applied in the functional prediction for such a large number of sequences in the *trans*-PTs family automatically.

To increase the similarity of sequence alignment and improve the accuracy of the positioning of “floor” residues, a sequence similarity network (SSN) was applied to generate the gene clustering map with its advantage of an excellent globe view for thousands of sequences and a good consistency with the phylogenetic tree [20]. A total number of 7870 *trans*-PT sequences from the Structure–Function Linkage Database (SFLD) [21] were analyzed using SSN with an e-value cutoff of 1e^−50^, and the visualization of the clustering map was performed in Cytoscape [22]. According to the clustering map, the *trans*-PTs protein family could be divided into six predominant clusters with several tiny satellite groups (Figure 2A). All the *trans*-PT sequences whose structures were determined were assigned into the SSN map, and 11 representative protein sequences were finally selected as searching templates which covered 85% of total sequences (6684/7870) in the SFLD dataset (Figure 2A). Of these representative proteins, six were from Bacteria (PDB IDs: 2FOR, 3OYR, 3QQV, 3PKO, 1WMW, and 3Q2Q), four belonged to Eukarya (PDB IDs: 5E8H, 2E8W, 1FPS, and 3AQ0), and one was from Archaea (PDB ID: 1WY0) The product specificity of these searching templates ranged from FPP (C15) to PPP (≥C30), and their structural information was used in the coordinate location of “floor” residues. These selected searching templates improved the average sequence identity from 33.9% to 46.1% over 141 testing sequences. The final average sequence identity for all 7870 sequences was 42.4% (Appendix A).

Next, we tried to design an automatic algorithm for high-throughput functional prediction of the *trans*-PTs protein family. We firstly assigned a “blocking score” to each amino acid on the basis of its side-chain geometry volume and torsion capacity (Appendix A) [23]; next, we calculated the “blocking score” for each of the “floors”; finally, chain lengths of the final products were predicted on the basis of the “blocking score” following the rules of the “three floors” model (Figure 2B). The weighting factors for each position of “floor” residues were trained using a series of experimental data including the 11 searching templates and 27 site-mutated *Arabidopsis trans*-PTs sequences (Dataset 1) [10]. The resulting trained weighting factors indicate that the following: (1) for the first floor, site_1 and site_2 have a similar contribution to the “blocking score”, while site_3 has much less importance (1.1% contribution to the final “blocking score” of “floor 1”); (2) in the second floor, site_1, site_2, and site_3 contribute 20.0%, 55.3%, and 24.7% to the “blocking score”, respectively; (3) in the third floor, similar with the first floor, site_1 and site_2 contribute 36.5% and 62.9%, respectively, while site_3 accounts for 0.6% contribution in the “blocking score” (Appendix A). Next, a rough polyprenyl pyrophosphate elongation route could be traced on the basis of these statistics following the rule that a higher weighting factor of the “floor” site indicates a greater likelihood of the elongation prenyl group passing by it (Appendix A). Interestingly, the traced elongation route is consistent with a previous study, which used ligand docking or co-crystallization to determine the elongation route in *trans*-PTs [14,24].

Taken together, the final equation was designed as an automatic algorithm and developed as a web tool named PTS-Pre (http://124.70.187.228:1080/, accessed on 18 August 2022) for further testing and high-throughput functional predictions.

### 2.2. Validation of PTS-Pre Prediction Using Experimentally Determined trans-PTs Sequences

To date, there is only one method that has been published for predicting the function of *trans*-PTs, which used structural modeling and covalent ligand docking (referred to as MD-Pre, “Modeling Docking Prediction method”, in the latter part of this paper) to predict the chain length specificity [14]. Since our proposed PTS-Pre tool utilizes a different mechanism, it was interesting to compare these two methods using the same testing dataset which was used for the MD-Pre as the initial testing. A total of 74 experimentally determined *trans*-PTs protein sequences were selected in MD-Pre which resulted in correct chain-length prediction for 58% (43/74) [14]. The same testing sequences were used for functional prediction by PTS-Pre which finally resulted in correct prediction for 77% (57/74) (Figure 3A, detailed prediction results are shown in Appendix A). A more detailed comparison shows that 37 sequences were correctly predicted by both methods, 20 sequences were correctly predicted only by the PTS-Pre method, and six sequences were correctly predicted only by PTS-Pre (Appendix A). Interestingly, among the 20 sequences which were correctly predicted only by PTS-Pre, 11 of them had activities for synthesizing final products longer than C30, while the predicted results by MD-Pre were much shorter (Appendix A). This may be because the longer final products (compared with shorter chain length products) have more flexibility and complexity for MD-Pre prediction based on ligand docking. Nevertheless, the different prediction results by MD-Pre and PTS-Pre suggest that the mechanisms behind these two methods are dissimilar, and the prediction accuracy may be improved by combining these two methods in the future.

Considering that experimental assays may also have a bias in identifying the product outcomes of *trans*-PTs (for example, the product chain length profile may change when using different protein purification tags or ratios of IPP/allylic co-substrates), and the detection methods may also have various sensitivities, we extended the testing dataset with more experimentally verified sequences which were analyzed via different detection methods including radio-gas chromatography (radio-GC), radio-high-performance liquid chromatography (radio-HPLC), thin-layer chromatography (TLC), and HPLC–MS/MS (Appendix A). A testing dataset consisting of 147 sequences was generated and applied to the functional prediction using PTS-Pre. The prediction accuracy for the new dataset was 86% (121/141), and detailed results are shown in Appendix A. To explore the scope of this method, we further analyzed the results by classifying these sequences into three super kingdoms and calculated the prediction accuracy for each of them. There were nine sequences in Archaea, 84 sequences in Bacteria, 46 sequences in Eukarya, and two unclassified sequences. Among the three categories, the prediction accuracies for eukaryotic, archaeal, and bacterial sequences were 95% (44/46), 88% (8/9), and 80% (67/84), respectively (Figure 3B). These results are not surprising since the “three floors” model was proposed on the basis of crystal structures from *Arabidopsis*; thus, the eukaryotic sequences had the highest prediction accuracy. However, the 88% and 80% accuracies for Archaea and Bacteria suggest that the “three floors” model could also be applied to these two kingdoms. We further analyzed the wrongly predicted cases. As the incorrectly predicted number was too small in Archaea (1/9) and Eukarya (2/46), only bacterial cases (17/84) were taken into consideration. A more detailed classification by phylum revealed that the distribution of wrongly predicted sequences was mainly concentrated in specific phyla (Appendix A). The 84 bacterial sequences from the testing dataset could be divided into 11 phyla, with the incorrectly predicted cases found only in six of them. There was only one sequence each in Chlamydiae and Chlorobi, but neither of them was predicted correctly. More incorrect cases were found in Deinococcus-Thermus, Actinobacteria, Firmicutes, and Proteobacteria with prediction accuracies of 33% (1/3), 71% (10/14), 80% (12/15), and 88% (35/40), respectively (Appendix A). Taken together, 85% (17/20) of total incorrectly predicted cases come from these bacterial phyla, which suggests that the prediction accuracy may be closely related to the species.

Consequently, we assigned all the testing sequences into the SSN map and compared the distribution of incorrect and correct predictions. As shown in Figure 2A, the testing sequences were distributed in most of the predominant clusters and dispersed randomly in the map, which was compatible as a testing dataset. Consistent with our hypothesis, the incorrect prediction cases were enriched in tiny clusters near the searching templates 3Q2Q (*Corynebacterium glutamicum*, Bacteria), 3PKO (*Levilactobacillus brevis*, Bacteria), and 1WMW (*Thermus thermophilus*, Bacteria), while the correct prediction cases were enriched in six main clusters with the searching templates 2FOR (*Shigella flexneri*, Bacteria), 5E8H (*Arabidopsis thaliana*, Eukarya), 1WY0 (*Pyrococcus horikoshii*, Archaea), 2E8W (*Saccharomyces cerevisiae*, Eukarya), 3OYR (*Caulobacter vibrioides*, Bacteria), 3AQ0 (*A. thaliana*, Eukarya), 3QQV (*Corynebacterium glutamicum*, Bacteria), and 1FPS (*Gallus gallus*, Eukarya). We further calculated the prediction accuracy for each of the searching templates, and the results show that the searching templates 1FPS (95% accuracy, 19/20), 2FOR (93% accuracy, 41/44), and 3OYR (88% accuracy, 15/17) had extremely high accuracies, while 3Q2Q (60% accuracy, 3/5), 3PKO (0% accuracy, 0/3), and 1WMW (66% accuracy, 2/3) had lower accuracies (Figure 3C; the complete data are shown in Appendix A). The accuracy for 3PKO is unusual as none of the three sequences (GI 28377915, 116334218, and 29376566) which selected 3PKO as the best template were correctly predicted. All three testing sequences belong to Firmicutes and synthesize products with chain lengths longer than C30 [14]. However, PTS-Pre predicted their activities as GGPPSs (C20) because the “second floor” of these proteins is blocked by two residues with large side-chains (these three proteins have the same “second floor” residues: site_1 ‘Leu’, site_2 ‘Phe’, site_3 ‘Met’) although they can pass through the “first floor” (“first floor” residues: GI 28377915, site_1 ‘Ala’, site_2 ‘Thr’, site_3 ‘Ile’; GI 28377915, site_1 ‘Gly, site_2 ‘Thr’, site_3 ‘Ile’; GI 28377915, site_1 ‘Ala’, site_2 ‘Thr’, site_3 ‘Leu’) (Appendix A). The conflict between the experimental data and the “three floors” model may be explained by the side-chain orientation rearrangement of blocking residues, thus creating a new product chain elongation tunnel along the interface of the *trans*-PT homodimer. This new chain elongation tunnel has been suggested by several previous studies in long-chain PTs [13,14], but whether all chain elongation products of the long-chain PTs would go along this tunnel is still unclear. On the basis of our testing sequences prediction results, 72% (26/36) of long-chain PTs were correctly predicted, indicating that at least most of them still followed the “three floors” model.

We also generated a phylogenetic tree to compare the distribution of the predicted testing sequences (Appendix A). The 141 sequences were divided into five clades. Similar to the observation in the SSN map, the incorrect predictions were enriched in the early branch of the phylogenetic tree while the correct predictions were gathered in the late branch. Taking these results together, we conclude that the application of the “three floors” model can be extended from the plant Brassicaceae family to species across all three life kingdoms, and the prediction accuracy may relate to their original species, displaying the highest accuracy for Eukarya (95%) and lowest for Bacteria (80%).

### 2.3. Genome Mining of the trans-PT Gene Family Identified More GFPPSs beyond the Brassicaceae Clade

After validating the PTS-Pre by experimentally verified testing sequences, we further employed the PTS-Pre in functional annotations for *trans*-PTs from whole species. First, we used the PTS-Pre in the functional identification of *trans*-PT gene family in some specific species. Three species were randomly selected from Bacteria and Eukarya including *Streptomyces azureus* (Actinobacteria), *Tolypothrix campylonemoides* (Cyanobacteria), and tree cotton *Gossypium arboretum* (Viridiplantae) for the analysis. A total of 12 sequences were applied to PTS-Pre analysis; the corresponding genes were synthesized, expressed in *E. coli*, and subjected to in vitro biochemical assays. As shown in Table 1 and Figure 3D, most of the *trans*-PTs from *T. campylonemoides* and *G. arboreum* were correctly predicted (prediction accuracy 87.5%, 7/8) except for *Ga*GGPPSL-4 (GI XP_017649086) and *Tc*GGPPSL-2 (GI KAB8317749). *Ga*GGPPSL-4 had no activity detected, while *Tc*GGPPSL-2 was predicted as GGPPS (C20) but finally revealed as GPPS (C10). The prediction accuracy for *S. azureus* was low (50%, 1/2), consistent with the previous observations through SSN assignment and phylogenetic tree analysis that genes from Bacteria had the lowest prediction accuracies.

To our surprise, two new GFPPSs from *T. campylonemoides* (*Tc*GGPPSL-4) and *G. arboreum* (*Ga*GGPPSL-3) were predicted and experimentally verified. According to the previous study, plant GFPPSs were only identified in the woody plant *Leucosceptrum canum* and several Brassicaceae species [10,25,26]. The newly identified GFPPS in *G. arboreum* suggests that there would be more sesterterpene (C25) related genes in other plant species. Therefore, we employed the PTS-Pre tool in genome mining for searching potential GFPPSs using a previously reported dataset which contains 7870 *trans*-PTs sequences [21]. The final prediction results revealed the abundance of *trans*-PTs with different chain lengths, indicating that FPPS (C15) had the highest abundance (39% of total sequences), while GGPPS (C20) and PPPS (≥C30) covered 24% and 32% of total sequences, respectively, and only 5% of the sequences were predicted as GFPPS (C25) (Figure 3E). We further classified these GFPPS sequences by their species, and 45 sequences were identified belonging to the plant kingdom (Dataset2). Sixteen representative sequences of the potential plant GFPPSs were synthesized and tested using in vitro biochemical assays. Finally, nine of them were identified with GFPPS activity (Figure 4A and Appendix A, Table 2). Interestingly, all these newly identified GFPPSs were beyond the Brassicaceae clade. To explore the evolutionary distribution of GFPPSs in the plant kingdom, we generated a phylogenetic tree using all predicted plant GFPPS candidates including the nine newly identified GFPPSs. The unrooted phylogenetic tree shows that the nine newly found GFPPSs, similar to GFPPSs from the Brassicaceae clade, are also lineage-specific (Figure 4B). The nine genes could be divided into four clades, namely, Nostocaceae, Malvaceae, Musaceae, and Ranunculaceae. It is noteworthy that the Nostocaceae family (which is a kind of Cyanobacteria) does not belong to the Viridiplantae kingdom but has a close evolutionary relationship with it. In the phylogenetic tree, the five GFPPS clades were located at different blanches and clearly separated. Interestingly, all the predicted GFPPSs in Poaceae clade including proteins from *Sorghum bicolor* (XP_021311468), *Panicum hallii* (XP_025814700), and *Setaria italica* (RCV24163) were finally determined as GGPPS (Figure 4A,B). Taken together, these results suggest that the GFPPS genes may have been the result of convergent evolution and are more likely lineage-specific.

## 3. Discussion

Among the various natural products synthesized in all living organisms, terpenoids are the most chemically, structurally, and functionally diverse array of metabolites with numerous applications in many aspects of our daily life [27]. Despite their enormous diversity, terpenoid carbon skeletons are all derived from polyprenyl pyrophosphate substrates of various chain lengths, which are determined by the reaction catalyzed by prenyltransferases [8]. Structurally and evolutionarily distinct from the short-chain *cis*-prenyltransferases reported in limited species [28,29], short-chain *trans*-prenyltransferases provide precursors for the vast majority of commonly found terpenoids including mono-, sesqui-, di-, and triterpenes, thus determining the types of terpenoid metabolites synthesized in a specific species [6]. The chemodiversity of terpenoids is further expanded by TPS enzymes, which often display promiscuous substrate specificities from in vitro assays [30,31]. However, the genuine in vivo activities of TPS enzymes necessarily depend on the substrate pools they can use [32]. With the fast development of genome sequencing, a rising number of prenyltransferases have been predicted in public databases, but a substantial portion of them were incorrectly annotated simply according to sequence similarities. A reliable method for the functional prediction of *trans*-PTs is important for the investigation of terpenoid biosynthetic pathways and will benefit the discovery of hidden natural products.

On the basis of our previously established “three floors” model [10], an automatic algorithm was designed using 11 searching templates that covered 85% of 7840 *trans*-PT sequences in the SFLD database (Figure 2A). The final equation was developed as a web tool named PTS-Pre for the precise functional prediction of *trans*-PTs (Figure 2B). Our analyses show that PTS-Pre yielded an 86% high prediction accuracy for 141 experimentally characterized *trans*-PTs covering Archaea, Bacteria, and Eukarya (Figure 3C), and the accuracy was as high as 90% when the predicted function was within one C5 unit. Compared with MD-Pre, the only known functional prediction method for *trans*-PTs [14], which had 58% accuracy over 74 experimentally determined *trans*-PTs, PTS-Pre displayed a 77% high accuracy over the same testing sequences (Figure 3A, Appendix A). The method was then validated by characterizing randomly selected genes from three species ranging from Actinobacteria to Cyanobacteria and Viridiplantae, revealing 100%, 75%, and 50% accuracies, respectively (Table 1). It seemed that more primitive genes yielded lower prediction accuracies. The sequence similarity network and phylogenetic tree further suggested the prediction accuracy may be related to the gene’s evolutionary position in the tree with a lower accuracy closer to the root (Appendix A). To further explore the molecular mechanism behind these observations, we compared the protein structures between bacterial *trans*-PTs and the one from *Arabidopsis*. Three bacterial structures (PDB IDs: 5H9D from *Staphylococcus aureus*, 1WKZ from *Thermotoga maritima*, and 1WY0 from *Pyrococcus horikoshii*) were superposed with the structure of AtGGPPS11 (PDB ID: 5E8L). All these three proteins form homodimers, and the root-mean-square deviation (RMSD) is as low as 1.499 Å over 260 residues, displaying very similar overall structures (Appendix A). We further compared the product chain elongation tunnels and found that the three key α helices (α helix D, E, and F) are also similar except for some nonessential residues (Appendix A). Thus, the inconsistency with the “three floors” model is unlikely related to the monomer structural factors. Some previous work indicated that, in addition to the residues in the “three floors”, the two aspartates in FARM and SARM were identified as factors influencing the chain length specificity of products in the Gram-positive bacterium *Mycobacterium tuberculosis* [33]. Moreover, the interface of the *trans*-PT homodimer was also proposed to be a new product chain elongation tunnel, in that the blocking residues were rearranged and affected the profile of the final product [14]. These reasons may explain the incorrectly predicted cases for bacterial *trans*-PTs by PTS-Pre; however, further investigations are still needed to verify whether these factors are only specific to bacteria or also cover other species. It also should be noted that the PTS-Pre method was designed on the basis of homodimeric structures and validated only by in vitro assays. However, in some specific species, the *trans*-PTs are in an additional heterodimeric state which contains one large subunit (LSU) and one small subunit (SSU). The kinetic parameters and product profile of the LSU would be changed by the regulatory function of SSU [12]; thus, the predicted results could be different from the genuine products formed in vivo, which require further in vivo investigations case by case.

Compared with the most extensively studied mono-, sesqui-, di-, and triterpenes (C10, C15, C20, and C30, respectively), sesterterpenes (C25) constitute the rarest group of terpenoids discovered to date, with only about 1300 members being reported [34]. Despite the wide distribution of sesterterpenoids in bacteria, fungi, insects, and marine organisms [19], the corresponding genes and enzymes of plant-originated C25 compounds were only recently reported in limited families including Lamiaceae and Brassicaceae [10,25,26]. Our blind prediction and subsequent enzyme assays of randomly selected testing sequences from *Gossypium* identified a new plant GFPPS (Table 1 and Figure 4), suggesting that more GFPPS-related genes may exist in the plant kingdom. Indeed, further genome mining of the *trans*-PT gene family identified six additional GFPPSs from Malvaceae, Musaceae, and Ranunculaceae families (Table 2 and Figure 4A), which were evolutionarily separated (Figure 4B). Many sesterterpenoids identified in plants possess either highly oxygenated or complex cyclic skeletons and exhibit diverse biological functions such as anti-inflammatory, antimicrobial, and antifeedant bioactivities [19,34]. The fact that changing only a few key amino acids could alter the enzymatic characteristics of *trans*-PT enzymes suggests that plant GFPPSs independently evolved via gene duplication and neo-functionalization for the adaption of various selection stresses. By using PTS-Pre, the prediction of promising GFPPSs with high credibility in specific species provides the fundamental basis for the discovery of new sesterterpenes and opens up a great opportunity for uncovering the mystery of the distribution and evolutionary origin of sesterterpene biosynthesis in plants.

## 4. Materials and Methods

### 4.1. Sequence Similarity Network and Phylogenetic Analysis

The *trans*-PTs sequences used to generate the similarity network were obtained from the Structure–Function Linkage Database (SFLD) [21]. The Enzyme Similarity Tool (EFI-EST) was then used to compute and generate the all-against-all pairwise comparison, and the resulting network was visualized at an e-value cutoff of 1 × 10^−50^ in Cytoscape 3.7.2 [22]. In the networks, nodes represent sequences, and edges indicate the sequence similarities that have a BLAST e-value more significant than the cutoff value. A total of 141 experimentally identified *trans*-PTs amino-acid sequences were used to generate the neighbor-joining tree using MEGA11 software [35] and to draw the final maps using the iTOL online software [36].

### 4.2. Gene cloning, Expression, and Protein Purification

All the genes tested in this study were de novo synthesized and cloned into the pET28a vector under the control of the bacteriophage T7 gene promoter using *Nde*I and *Xho*I. Recombinant proteins were expressed in *E. coli* BL21(DE3) strains. Cells were grown at 37 °C to reach an OD600 of approximately 0.5, and then induced by adding isopropyl β-d-1-thiogalactopyranoside (IPTG) to a final concentration of 0.5 mM. After incubating for 16 h at 16 °C, the cells were harvested by centrifugation (5000× *g* for 10 min at 4 °C). The recombinant His-tagged and GST-tagged proteins were purified using a Ni^2+^-nitrilotriacetic acid (Ni-NTA) affinity column and Glutathione–Sepharose column according to the manufacturer’s instructions. Purified proteins were desalted, and the buffer of proteins was exchanged to reaction buffer (250 mM MOPS, 100 mM MgCl_2_, pH 7.0, 20 mM DTT) by PD SpinTrap G-25 (GE Healthcare Biosciences, Uppasla, Sweden). The final proteins were identified by sodium dodecyl sulfate (SDS) 12% polyacrylamide gel electrophoresis (PAGE) and stained with Coomassie brilliant blue (Appendix A).

### 4.3. In Vitro trans-Prenyltransferase Activity Assays

The enzyme activity assays were carried out as described in [10]. A total of 10 μg of protein was added to the 100 μL enzymatic reaction system (250 mM MOPS, 100 mM MgCl2, pH 7.0, 20 mM DTT), using [1–14C]-IPP (55 mCi/mmol, American Radiolabeled Chemicals) and DMAPP as substrates. After the enzyme activity assays (at 30 °C for 1 h), 500 μL lysis buffer of 0.2 M Tris-HCl (pH 9.0) containing one unit of potato apyrase (Sigma-Aldrich) and 1 μL of bovine intestine alkaline phosphatase (TaKaRa) was added to the enzymatic reaction system, followed by incubation at 30 °C overnight. After incubation, 600 μL of hexane was added to the reaction system to extract the polyprenyl alcohols, then vortexed for 2 min, and centrifuged at 12,000× *g* for 5 min. Next, 500 μL hexane extracts were concentrated to 50 μL under nitrogen gas, and then separated using a reverse-phase (C18 silica gel-60 matrix, F254S) thin-layer chromatography (TLC) plate (Merck, Darmstadt, Germany). Acetone/water (9:1 *v*/*v*) was used as the mobile phase. The separated polyprenyl alcohols were detected using the Typhoon Trio Variable Mode Imager (GE Healthcare Biosciences).

### 4.4. Computational Function Annotation Design of PTS-Pre

PTS-Pre is a hierarchical approach to predict the function of *trans*-PTs, consisting of three consecutive steps: (1) searching for the best template for a target gene and locating the six “floor” residues in the sequence, (2) calculating the “blocking score” for each of the “floors”, and (3) determining the final products according to the “blocking score” and the “three floors” model. Starting from the query amino-acid sequences, we firstly run NWAlign to create multiple sequence alignments with the target sequence and the 11 searching template sequences (PDB IDs: 2FOR, 3OYR, 3QQV, 3PKO, 1WMW, 3Q2Q, 5E8H, 2E8W, 1FPS, 3AQ0, and 1WY0). The searching sequence with the highest similarity is used as the template for locating the six “floor” residues in the target sequence before forwarding the position information and the type of amino acid to the second step. Next, the “blocking scores” for each of the “floors” are calculated using an equation with three weighting factors for each of the three sites in the “floor” (Appendix A). Finally, the “blocking scores” are compared with the threshold to determine the final products for the target sequences. For example, if the first floor’s blocking score is larger than the first floor’s threshold, then the final products are returned as FPP (C15); otherwise, the second run of the comparison proceeds for the second floor (Figure 2B).

## Figures and Tables

**Figure 1 ijms-23-09471-f001:**
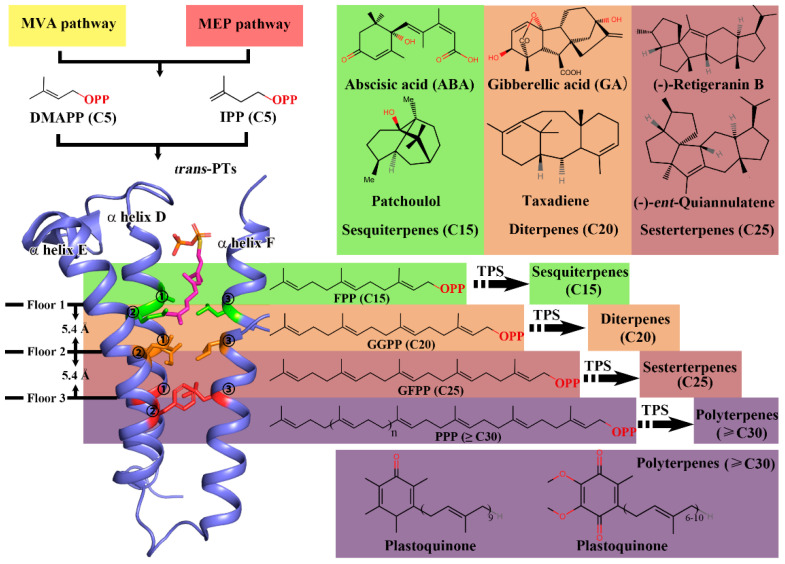
The “three floors” model and scheme of terpenoid biosynthetic pathway. The “three floors” model is illustrated using AtGGPPS11. For clarity, a substrate FPP (magenta sticks) is modeled into the structure using 3WJN as the reference model. The side-chains of residues from the “floors” are shown with sticks and colored in green (“first floor”), orange (“second floor”), and red (“third floor”). Dimethylallyl pyrophosphate (DMAPP) and isopentenyl pyrophosphate (IPP) from MVA/MEP pathways are used as substrates by *trans*-PTs to generate polyprenyl pyrophosphates with different chain lengths which are determined by the “three floors” model. These products from *trans*-PTs are the precursors of terpene synthases (TPSs) and can be further utilized to generate various terpenoids including sesquiterpenes (C15), diterpenes (C20), sesterterpenes (C25), and polyterpenes (≥C30).

**Figure 2 ijms-23-09471-f002:**
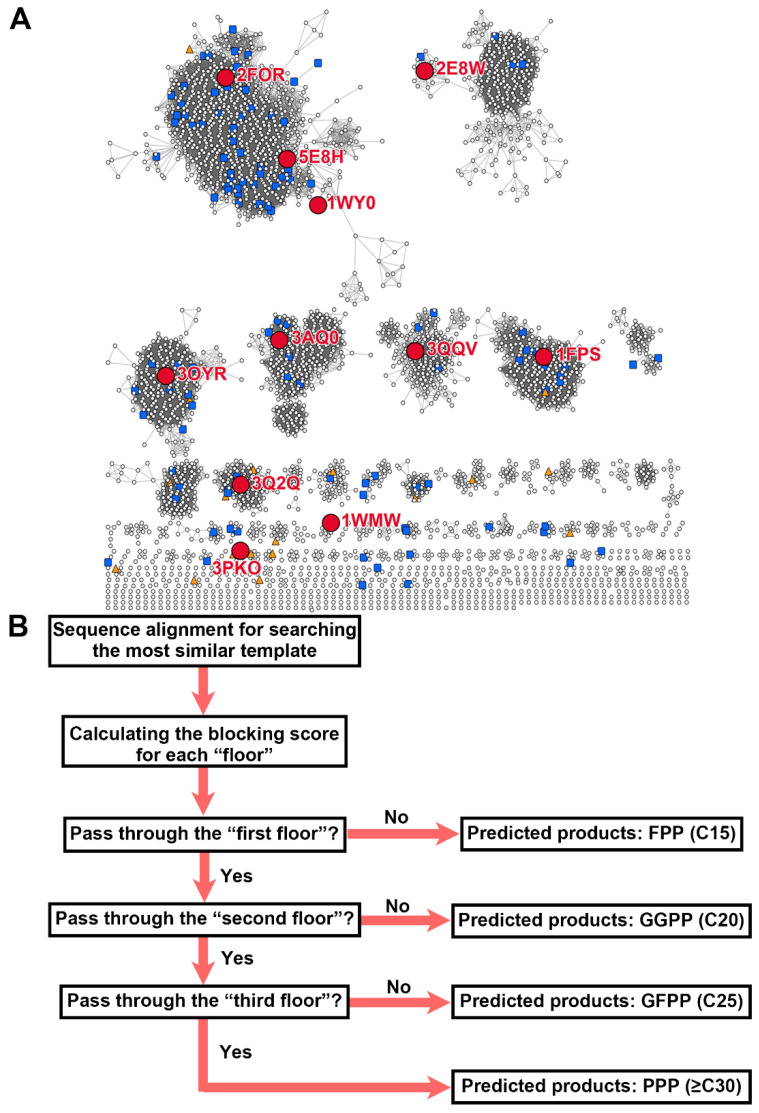
Sequence similarity network map of the *trans*-PT family and flowchart of PTS-Pre tool. (**A**) Sequence similarity network of the *trans*-PTs family with BLAST e-value cutoff set to 1e^−50^. Eleven representative *trans*-PTs (PDB IDs: 2FOR, 5E8H, 1WY0, 2E8W, 3OYR, 3AQ0, 3QQV, 1FPS, 3Q2Q, 3PKO, and 1WMW) are marked as large red circles. The incorrect and correct prediction cases from the experimentally determined testing dataset are marked as orange triangles and blue squares, respectively. (**B**) The general strategy followed by PTS-Pre for the functional prediction of *trans*-PTS. First, the target protein is aligned with each of the searching template sequences, and the best template with the highest sequence identity is selected as a reference for locating the six “floor” residues. Afterward, PTS-Pre calculates the “blocking score” for each “floor” on the basis of the types of amino acids and the parameter weights for each site. Finally, according to the “blocking scores”, PTS-Pre predicts the products’ chain length of target enzymes.

**Figure 3 ijms-23-09471-f003:**
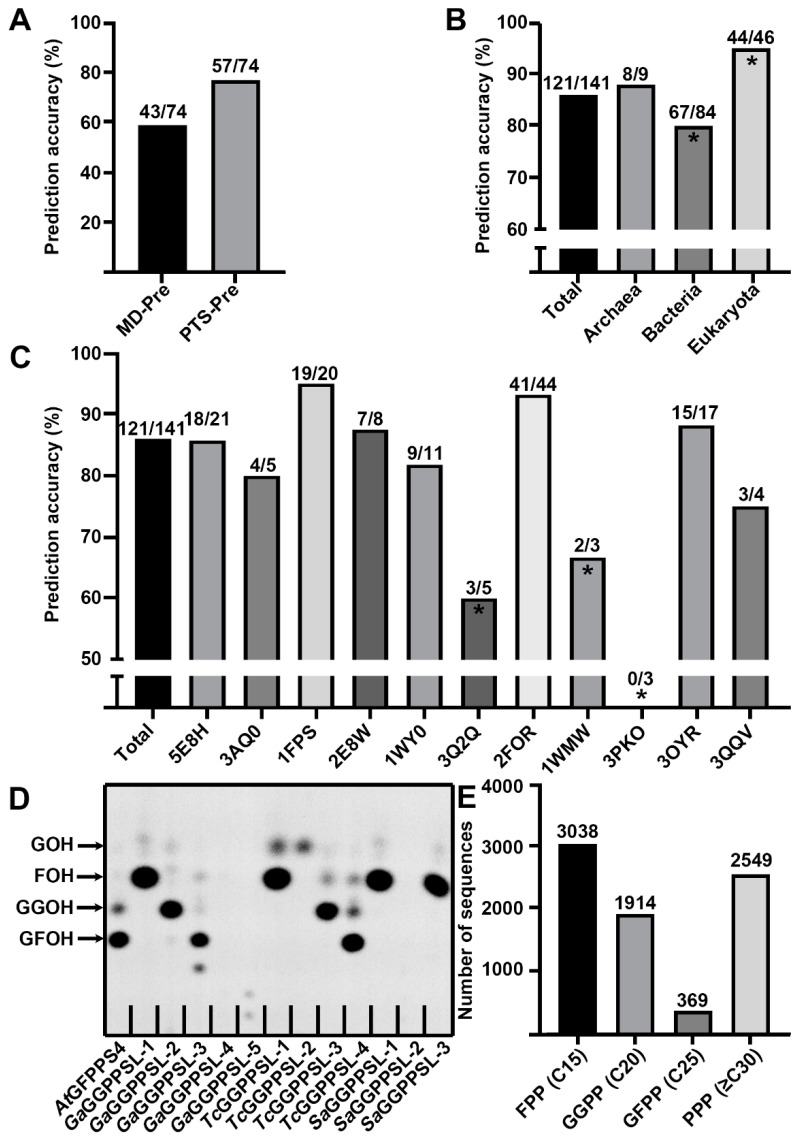
The prediction and verification results of PTS-Pre. (**A**) Comparison of prediction accuracies between MD-Pre and PTS-Pre over 74 testing sequences. (**B**) Comparison of prediction accuracies across three life kingdoms. Panels with comparatively higher or lower accuracy are marked with asterisks. (**C**) Comparison of prediction accuracies using different searching templates. Panels with comparatively higher or lower accuracy are marked with asterisks. In (**A**–**C**), the quantities of correctly predicted and total cases are labeled on the top of each panel. (**D**) In vitro activity of predicted *trans*-PTs from *Streptomyces azureus*, *Tolypothrix campylonemoides*, and *Gossypium arboreum*. All predicted *trans*-PTs were characterized using DMAPP and [1–14C]-IPP as co-substrates; the products were analyzed by TLC. AtGFPPS4 was used as a positive control. (**E**) The distribution of predicted products in the *trans*-PTs family.

**Figure 4 ijms-23-09471-f004:**
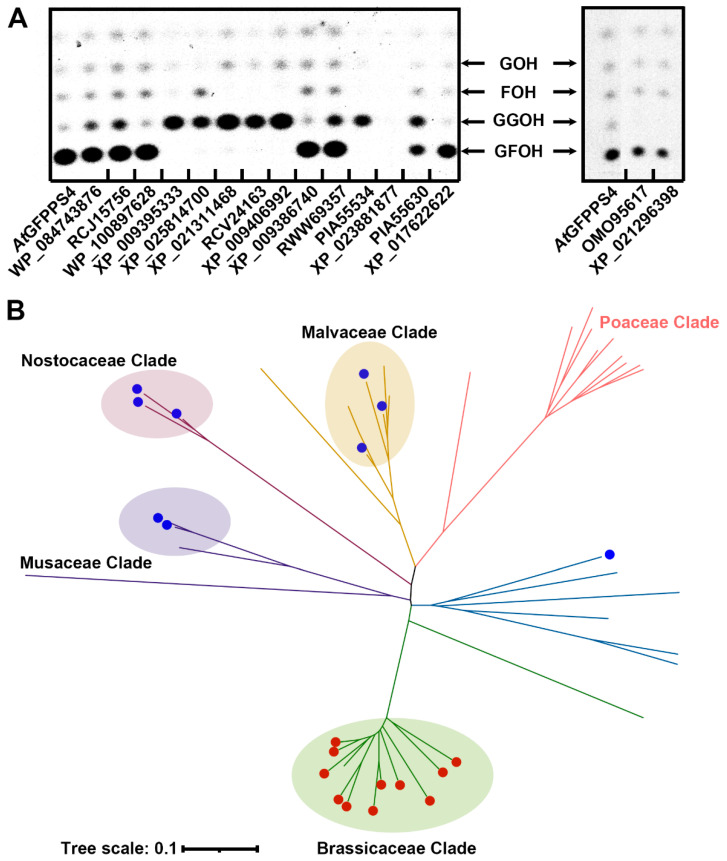
Identification and phylogenetic analysis of GFPPS homologs in plants. (**A**) In vitro enzymatic activities of predicted GFPPS candidates. All the predicted *trans*-PTs enzymes were characterized using DMAPP and [1–14C]-IPP as co-substrates; the products were analyzed using TLC. AtGFPPS4 was used as a positive control. (**B**) Phylogenetic analysis of potential GFPPSs predicted using PTS-Pre with experimentally identified GFPPSs from panel A. The GFPPSs from *Arabidopsis thaliana* are highlighted by red circles, and the GFPPSs identified in this study are highlighted by blue circles. The branches of the phylogenetic tree are colored according to different clades; Musaceae, Nostocaceae, Malvaceae, Poaceae, and Brassicaceae are colored with purple, firebrick red, yellow, red, and green, respectively.

**Table 1 ijms-23-09471-t001:** The predicted and experimentally verified *trans*-PTs from three randomly selected species.

Name	Protein ID	Best Template	Sequence Identity (%)	PTS-Pre Predicted	Experimentally Determined
*Ga*GGPPSL-1	XP_017641950	1FPS	45.4	C15	C15
*Ga*GGPPSL-2	XP_017620811	5E8H	67.5	C20	C20
*Ga*GGPPSL-3	XP_017622622	5E8H	59.9	C25	C25
*Ga*GGPPSL-4	XP_017649086	3AQ0	71.8	≥C30	NA
*Ga*GGPPSL-5	XP_017640526	3AQ0	74.7	≥C30	≥C30
*Tc*GGPPSL-1	KAB8318528	2FOR	49.8	C15	C15
*Tc*GGPPSL-2	KAB8317749	3AQ0	39.4	C20	C10
*Tc*GGPPSL-3	KAB8318557	5E8H	50.9	C20	C20
*Tc*GGPPSL-4	KAB8315220	5E8H	51.0	C20/C25	C25
*Sa*GGPPSL-1	WP_059416577	2FOR	36.1	C15	C15
*Sa*GGPPSL-2	WP_059424690	3Q2Q	43.5	C20	NA
*Sa*GGPPSL-3	WP_078945674	3OYR	35.2	C20/C25	C15

*Ga*GGPPSL-1–5, *Tc*GGPPSL-1–4, and *Sa*GGPPSL-1–3 are from *Gossypium arboreum*, *Tolypothrix campylonemoides*, and *Streptomyces azureus*, respectively; NA, no activity was detected.

**Table 2 ijms-23-09471-t002:** The prediction and identification of potential GFPPSs in plants.

Protein ID	Organism	PTS-Pre Predicted	Experimentally Determined
WP_084743876	*Tolypothrix campylonemoides* *	C20/C25	C25
RCJ15756	*Nostoc* sp. *ATCC 43529* *	C20/C25	C25
WP_100897628	*Nostoc flagelliforme* *	C20/C25	C25
XP_009395333	*Musa acuminata*	C20/C25	C20
XP_025814700	*Panicum hallii*	C25	C20
XP_021311468	*Sorghum bicolor*	C25	C20
RCV24163	*Setaria italica*	C25	C20
XP_009406992	*Musa acuminata*	C25	C20
XP_009386740	*Musa acuminata*	C25	C25
RWW69357	*Ensete ventricosum*	C25	C25
PIA55534	*Aquilegia coerulea*	C25	C20
XP_023881877	*Quercus suber*	C25	NA
PIA55630	*Aquilegia coerulea*	C25	C25
XP_017622622	*Gossypium arboreum*	C25	C25
OMO95617	*Corchorus olitorius*	C25	C25
XP_021296398	*Herrania umbratica*	C25	C25

* Species belonging to Cyanobacteria.

## Data Availability

The webserver and dataset are available at http://124.70.187.228:1080/, accessed on 18 August 2022.

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
