# Peer review of "Functional Prediction of trans-Prenyltransferases Reveals the Distribution of GFPPSs in Species beyond the Brassicaceae Clade"

_ijms, 2022, doi:10.3390/ijms23169471_

Round 1
Reviewer 1 Report
The manuscript describes the application of the model for predicting products of the IPP/DAMPP condensation catalyzed by prenyltransferases (PTs) based on their sequence/structure relationships. The manuscript is well written, and the presented evidence and the data interpretation are quite convincing.
The proposed model of inferring reaction products from the structural features of the catalytic center obviously has many merits for PTs natively occurring as homodimers and shows good accuracy in predicting the outcome of the reaction. For that reason alone, following some minor language corrections, the manuscript certainly deserves publication in IJMS.
However, the authors seem to entirely neglect the presence of heterodimeric PTs in plants. Even the enzyme with the structure shown in Figure 1, AtGGPPS11, is known to exist as a heterodimer with a smaller sub-unit regulating not only the kinetic parameters of the reaction but also the product profile. Therefore, even though the AtGGPPS11, based on its structural features, can catalyze the synthesis of GGPP (diterpene), the actual product of the reaction is a mixture of GGPP and GPP (monoterpene).
Because of this omission, the current model proposed by the authors significantly oversimplifies the possible outcome of the PT-catalyzed reactions in plants. To mitigate this, the authors should consider augmenting the model for the possibility of heterodimer formation. Perhaps PT sequences could be scanned and flagged for conservative motifs responsible for sub-units interactions.
Similarly, if the given protein can form heterodimers, the results of activity assays carried out without a regulatory subunit may not reflect the actual product of the reaction. Therefore, they may not be suitable for validating the current model and perhaps should be removed from the dataset.
At the very least, the authors should warn the readers and the users of the model about the possibility of obtaining inaccurate prediction results due to heterodimer formation.
Reviewer 2 Report
Overall, the research addresses the question of how to identify trans-preyltransferases which is the first branch for generation of terpenoids. Research in this space is important to identify and generate new natural products. The authors present an empirical decision “graph” in the form of a web tool based in their experiences with trans-PTs. They test their algorithms and identifies new potential tPTs and test these experimentally with 9 of them as tPS. The paper is well written - there are some formatting issues on some of the tables but otherwise fine. Major suggestion: Identification of new enzymes should be performed using large language models and for a sustainable prediction it would be good to train some neural networks for these predictions. Also the advent of alphafold - it would be good if there would be a link to AF prediction in the web page - also maybe having the structure would be the CX predictions better (e.g., C20 vs C25 etc). Minor questions/comments to the paper: Why is it urgently needed? Table 2 -> C20/C25 why is many switched?Author Response
Please see the attachment.
